# Intraovarian Injection of Reconstituted Lyophilized Growth-Promoting Factor Extracted from Horse Blood Platelets (L-GF^equina^) Increases Oocytes Recovery and In Vitro Embryo Production in Holstein Cows

**DOI:** 10.3390/ani12192618

**Published:** 2022-09-29

**Authors:** Silviu-Ionuț Borş, Dan-Lucian Dascălu, Alina Borş, Hossam M. Fahmy, Omaima M. Kandil, Ahmed Sabry S. Abdoon

**Affiliations:** 1Research and Development Station for Cattle Breeding Dancu, Holboca, 707252 Iaşi, Romania; 2Department of Public Health, Faculty of Veterinary Medicine, University of Life Sciences “Ion Ionescu de la Brad” Iaşi, 700490 Iaşi, Romania; 3Clinical Laboratory and Blood Bank Department, Faculty of Medicine, Ain Shams University, Cairo 1181, Egypt; 4Department of Animal Reproduction and Artificial Insemination, Veterinary Research Institute, National Research Centre, Dokki, Cairo 12622, Egypt

**Keywords:** blastocysts production, L-GF^equina^, follicular development, Holstein cow, OPU-IVEP

## Abstract

**Simple Summary:**

Lyophilized horse-platelet-derived growth factor (L-GF^equina^) is a brand-new, improved platelet-rich protein growth factor. It has been successfully used to treat a number of inflammatory and degenerative musculoskeletal diseases in several disciplines of regenerative medicine. Our study aimed to evaluate the possibility of using an intraovarian injection of L-GF^equina^ in the assisted reproduction of Holstein cows. L-GF^equina^ employs allogenic, pathogen-free platelets from horses that have been successfully used in other species to prevent changes in technical platelet specifications caused by the centrifugation process and time, as well as the integrity and quality of autologous platelets. The number of ovum pick-up (OPU) sessions per cycle, the growth of medium- and large-sized follicles, the number of COCs produced per cow per session, and the formation of blastocysts during OPU-IVEP in eCG stimulated Holstein cows were all boosted in this study by an intraovarian injection of L-GF^equina^.

**Abstract:**

The purpose of this study was to determine the impact of intraovarian injections of a reconstituted lyophilized growth-promoting factor extracted from horse blood platelets (L-GF^equina^) on the number of ovarian follicles, the recovery of cumulus–oocyte complexes (COCs), and embryo development to the blastocyst stage in Holstein cows. Thus, 12 Holstein cows were assigned to three protocols. According to the number of punctured follicles in protocol 1, ovum pick-up (OPU) was conducted on days 6 and 14 of the cycle (day 0 = estrus). In protocol 2, every large follicle (more than 7 mm) was removed, and 1 mL of L-GF^equina^ was intraovarian injected (day 0). Two days later, equine chorionic gonadotropin (eCG) was administered, and OPU sessions were conducted on days 6, 10, and 14. The same ovarian stimulation procedure as that in protocol 2 was performed in protocol 3, except that equine L-GF^equina^ was not supplied. OPU was carried out on days 6 and 10 of the cycle. The results indicate that the intraovarian injection of L-GF^equina^ significantly (*p* < 0.05) increased the number of OPU sessions per cycle, the recovery of cumulus–oocyte complexes (COCs), and the production of blastocysts. In conclusion, an intraovarian injection of L-GF^equina^ can improves OPU-IVEP results in Holstein cows.

## 1. Introduction

The combination of assisted reproductive technologies (ART), such as ovum pick-up (OPU), in vitro embryo production (IVEP), and genetic selection, can rapidly enhance the success of the dairy and beef cattle industries. OPU and IVEP have become the most widely used tools for IVEP technology in the research and commercial areas. However, the success of these techniques depends on the quality and quantity of aspirated cumulus–oocyte complexes (COCs) per session. Therefore, an effective protocol for OPU-IVEP should be established. In the dairy industry, the application of OPU and IVEP allows the production of a large number of embryos from elite donors in a short period [1], and it accounts for 39.5% of the total embryo production worldwide (1,156,422 embryos) [2]. Despite the huge progress in this field, the OPU and IVEP procedures are not fully optimized, and their efficiency is still relatively low. In this respect, the success of OPU depends on many factors, such as variation in antral follicle count (AFC) among individuals, cattle breeds [3], and the stage of follicular growth at which the OPU session is performed [4]. By using various schemes of twice-weekly OPU, less than three COCs from non-stimulated cows were aspirated [5], but more than nine COCs were recovered from cows that had been stimulated with gonadotropin [6]. Additionally, compared to continuous OPU sessions, which have been shown to affect the rate of oocyte recovery and the estrous cycle by altering the endocrine function and the mechanisms of follicular growth, leading to irregular estrous intervals [7] or the absence of estrus [5], OPU sessions repeated at 3-to-4 day intervals produced a higher recovery rate and produced more COCs [8,9]. Therefore, there is still a need for a new OPU protocol that can support the repeated use of donors for the development of a large number of antral follicles and improve in vitro blastocyst development without altering the ovarian structure or its functions.

During the last decade, the work on platelet-rich plasma (PRP) has attracted attention in many fields of veterinary medicine. Platelet-rich plasma has been used clinically in animals for its healing properties due to the increased concentrations of autologous growth factors [10]. Platelet-derived growth factor (PDGF), transforming growth factor (TGF), vascular endothelial growth factor (VEGF), epidermal growth factor (EGF), basic fibroblast growth factor (bFGF), and insulin-like growth factor (IGF-1) are some of the growth factors released from platelets that are involved in the healing process [11]. Platelets also release other factors that promote tissue repair and significantly impact the reactivity of vascular and other blood cells in angiogenesis and inflammation [12]. Additionally, PRP contains cytokines such as platelet factor 4 (PF4) and CD40 ligand (CD40L) [13]. Treatment with PRP is a valuable therapy for improving the embryo recovery in superovulating cows [14]. Furthermore, it has been used to increase fertility in cows with ovarian hypofunction [15] and to increase the pregnancy rate in repeat breeding of pure Arabian mares [16]. Platelet-rich plasma could modulate the inflammatory and immune activation processes and potentially minimize inflammation-associated tissue damage [17]. Therefore, in donors undergoing repeated OPU, PRP therapy could be an effective alternative strategy to reduce the detrimental effects of inflammation and subsequent fibrosis on ovarian function.

For PRP preparations, various commercial systems are available. Due to changes in technical platelet specifications, relating to the centrifugation process and time, as well as the integrity and quality of platelets, there may be significant variations in the final platelet concentrate [18].

In place of autologous platelets, L-GF^equina^ uses allogenic, equine pathogen-free platelets which have been used with success in another species. It is a revolutionary, advanced, standardized, and refined form of platelet growth factor [19]. To the best of our knowledge, this is the first study to investigate the effect of intraovarian injection of L-GF^equina^ into another species. This work aims to evaluate the effect of priming with reconstituted lyophilized L-GF^equina^ before eCG ovarian stimulation on follicular development, oocytes recovery, and in vitro embryo production in Holstein cows.

## 2. Materials and Methods

### 2.1. Experimental Animals

This study was performed at the Research and Development Station for Cattle Breeding Dancu-Iasi, Romania, which has a population of 330 dairy cows of the Holstein breed. The animals used within this study (*n* = 12) were healthy multiparous cows, with an average milk production of 25.1 ± 6.8 kg milk/day, at 168.9 ± 46.7 days in milk, aged between 4–5 years, with a body weight ranging from 550 to 600 kg, and body condition scores of 3–3.25 (on a 0-to-5-point scale) [20]. The cows were housed in free-stall barns with concrete, covered with mattresses, and fed a Total Mixed Ration (TMR) two times/day with ad libitum access to water.

### 2.2. Lyophilized L-GF^equina^

Dr. Hossam M. Fahmy, Professor of Laboratory and Transfusion Medicine, Ain Shams Medical School, Cairo, Egypt, devised a patented procedure for producing L-GF^equina^ (Code number: WO2018091713). PRP was extracted using an apheresis device. The Mirasol system (Terumo BCT, Lakewood, CO, USA) was used to treat the collected PRP with UV/riboflavin for pathogen inactivation and viral reduction. Following this, platelets were stimulated in vitro with thrombin and calcium chloride to release growth factors from the alpha granules. Among the growth factors, transforming growth factorβ, platelet derived-growth factor, fibroblast growth factor, IGF-1, and thrombasthenia were present. Following purification and separation from cellular waste and fibrin clot, the released growth factors were subjected to the solvent/detergent procedure for viral inactivation, followed by sterile filtration. Therefore, it was considered that any potential microbial contamination had been removed. Then, using aseptic techniques, the purified, sterile growth factors were given out in sterile vials. Each vial’s growth factor concentration was equivalent to the amount found in 20 mL of whole blood from a healthy donor, guaranteeing consistency in growth factor concentrations and clinical effects both within and between batches. In order to reconstitute the solution, 10 mL of sterile saline solution was added to a bottle of L-GF^equina^.

### 2.3. Assignment of Groups and Protocol of Administration

The cows from the first group (Prot 1, *n* = 6) were gynecologically examined using transrectal ultrasonography (iScan 2, Draminski S.A., Olsztyn, Poland) for the detection of the corpus luteum (CL) and then synchronized into estrus by injecting 500 µg cloprostenol (PGF2α, PGF Veyx forte, Veyx-Pharma GmbH, Schwarzenborn, Germany). All cows showed estrus (day 0) within 2–3 days after PGF2α administration. Transrectal ultrasonography (iScan 2, Draminski S.A., Olsztyn, Poland) was performed every 2 days, and according to the presence of several follicles that could be punctured, OPU sessions were applied on days 6 and 14, after detectable estrus (Figure 1, Prot 1). After a 3-week interval, the cows from Prot 1 were included in the second protocol (Prot 2, *n* = 6). In Prot 2, all large follicles (>7 mm) were removed (LFR) by OPU followed by an intraovarian (subcortical) injection of 1 mL reconstituted L-GF^equina^/ovary and, then, 2 days later by i.m. injection of 2000 IU equine chorionic gonadotropin (eCG, Folligon, MSD Animal Health, Kenilworth, NJ, USA). The intraovarian injection of 1 mL reconstituted L-GF^equina^ was performed by adapting a 5 mL syringe to the OPU device. Transrectal ultrasonography was performed every 2 days, and OPU sessions were conducted on days 6, 10, and 14 due to the presence of a large number of follicles at ultrasound examination (Figure 1, Protocol 2). For the second group (Prot 3, *n* = 6), 2 or 3 days after spontaneous estrus, all follicles > 7 mm were removed (LFR) by OPU, followed 2 days later by i.m. injection of 2000 IU eCG. Transrectal ultrasound examination was performed every 2 days, and according to the presence of several follicles that could be punctured, OPU sessions were conducted on days 6 and 10 (Figure 1, Prot 3). Due to the lack of any medium or large follicles during the examination of the ultrasonography every two days, one cow was removed from Prot 3 on day 6.

### 2.4. Ovum Pick-Up

The cows were restrained in a stanchion, which allowed minimal movement. They were prepared for OPU by being administered with 100 µg/kg BW xylazine HCL (Narcoxyl 2, MSD Animal Health, Boxmeer, Netherlands) i.m., followed 10 min later by 7 mL epidural anesthesia of 2% procaine hydrochloride (procaine 2%, Romvac, Voluntari, Romania). Transvaginal ultrasound-guided oocyte collection was performed using an ultrasound scanner (Aloka–SSD Prosound 2 scanner, Hitachi Medical System, Tokyo, Japan) with a convex-sector probe at 5 MHz attached to an aspiration pump (Rocket medical, Watford, UK) and ovum pick-up needle (COVA Needle “type A”, 17 G × 500 mm, Minitube GmbH, Tiefenbach, Germany) guidance system. For oocyte aspiration, a vacuum of 100 mmHg was used. All follicles > 3 mm were aspirated, and special attention was given to avoid partial aspiration or completely missing a follicle. All visible follicles were quantified and classified according to their diameters (small follicles: 2 to 3 mm, medium follicles: 4 and 6 mm, and large follicles: more than 7 mm [21]). COCs were harvested from ovaries by OPU in a 50 mL Falcon (Minitube GmbH, Hauptstraße 4184184 Tiefenbach, Germany) tube containing an OPU medium (IVF Bioscience, Bickland Industrial Park, Falmouth, UK). The same veterinarian aspirated the follicular fluid, collected it in a 50 mL Falcon tube, and left it to settle for 15 min in a water bath at 37 °C.

### 2.5. In Vitro Embryo Production Protocol

The immature oocytes were transferred using a 1.5 mL Pasteur pipettes from the bottom of the Falcon tube to a 9 cm sterile culture dish (Greiner Bio-One GmbH, Maybachstraße 272636 Frickenhausen, Germany) containing 10 mL of OPU medium (IVF Bioscience, Bickland Industrial Park, Falmouth, UK). The selection of the good-quality oocytes based on the surrounding cumulus cell layers and the homogeneity of the ooplasm was performed with a stereo microscope (Olympus SZ-51, Tokyo, Japan). After classification into different grades, according to their morphology, the oocytes were washed three times in wash medium (Wash Oocyte and Embryo Wash Medium, IVF Bioscience, Bickland Industrial Park, Falmouth, UK), transferred to 500 μL of maturation medium for oocytes (BO-HEPES-IVM™ HEPES-Buffered Oocyte Maturation Medium, IVF Bioscience, Bickland Industrial Park, Falmouth, UK), and kept in an incubator at 38.5 °C under 5% CO_2_ in humidified air for 24 h. After maturation, the oocytes were washed at least three times in a BO-IVF medium and placed in a 90 μL micro-drop of BO-IVF™ Fertilisation Medium (IVF Bioscience, Bickland Industrial Park, Falmouth, UK) covered with mineral oil (Oil for Medium Overlay, IVF Bioscience, Bickland Industrial Park, Falmouth, UK). Before use, the 90 μL micro-drop of BO-IVF™ Fertilisation Medium covered with mineral oil was equilibrated at 38.5 °C in a CO_2_ incubator under 5% CO_2_ in humidified air for 2 h. Frozen–thawed spermatozoa from a sire of proven fertility were used for the in vitro fertilization. Semen straws were thawed at 37 °C for 30 s, and then the frozen–thawed semen was centrifugated two times in BO-SemenPrep™ Semen Preparation Medium (IVF Bioscience, Bickland Industrial Park, Falmouth, UK). After centrifugation, the surplus of Bo-SemenPrep medium was aspirated, and the semen pellet was mixed with the remaining 200 μL Bo-SemenPrep. Approximately 1 × 10^6^ sperm/mL (10 μL of sperm solution) was also submitted to the fertilization 90 μL micro-drops, and then the gametes were co-incubated for 18–22 h at 38.5 °C in a CO_2_ incubator under 5% CO_2_ in humidified air. For the cultivation of presumptive zygotes, cumulus cells were removed and washed three times in washing medium (Wash Oocyte and Embryo Wash Medium, IVF Bioscience, Bickland Industrial Park, Falmouth, UK) to remove the remaining cumulus cells and cultured in 100 μL BO-IVCTM Embryo Culture Medium (IVF Bioscience, Bickland Industrial Park, Falmouth, UK) under mineral oil (Oil for Medium Overlay, IVF Bioscience, Bickland Industrial Park, Falmouth, UK) at 38.5 °C in a CO_2_ incubator under 5% CO_2_ in humidified air. Before use, the medium was equilibrated at 38.5 °C in a CO_2_ incubator under 5% CO_2_ in humidified air for 2 h. For all IVEP procedures (IVM, IVF, and IVC), we used a 35/10 mm cell culture dish, Vents, Cellstar^®^ TC, sterile (Greiner Bio-One GmbH, Maybachstraße 272636 Frickenhausen, Germany). The results of the IVEP procedure were evaluated on days 7 and 8 (the day of fertilization was considered day 0).

### 2.6. Statistical Analysis

A comparison of data from a single animal over time and then a comparison of data between OPU sessions were conducted using two-way ANOVA [22]. A chi-square model was used in the final analysis after the removal of non-significant interactions. For multiple comparisons, Tukey tests were used. The differences were significant if *p* ≤ 0.05. The obtained data are presented in the text as mean ± standard deviation.

## 3. Results

In this study, the OPU sessions were carried out on days 6 and 14 for protocol 1; days 6, 10, and 14 for protocol 2; and days 6 and 10 for protocol 3, in accordance with the presence of numerous follicles that can be punctured. (Figure 2).

Table 1 presents the effect of the OPU protocols on follicular development and blastocyst production. On Day 6, the total number of follicles was higher (*p* < 0.05) in protocols 2 and 3 when compared with protocol 1. At the same time, the number of 3–4 mm diameter follicles was higher (*p* < 0.05) in protocol 3 than in protocols 2 and 1. The number of 5–6 mm diameter follicles was higher (*p* < 0.05) in protocols 2 and 3 than in protocol 1. Additionally, the number of >7 mm diameter follicles was significantly (*p* < 0.05) higher in protocol 2 than in protocol 3, and both were higher (*p* < 0.05) than in protocol 1. Additionally, protocol 3 had much more COCs recovered and a higher recovery rate than protocol 2 (*p* < 0.05), and both protocols 2 and 3 had significantly more COCs recovered and a higher recovery rate than protocol 1 (*p* < 0.05). Blastocyst production was higher (*p* < 0.05) in protocols 2 and 3 than in protocol 1. On day 10, the comparison between L-GF^equina^ priming before eCG superstimulation (protocol 2) and eCG superstimulation alone (protocol 3) revealed that priming with L-GF^equina^ before eCG injection significantly (*p* < 0.05) increased the total number of follicles, the number of 5–6 mm and >7 mm diameter follicles, the number of COCs, the recovery rate, and blastocyst production when compared with eCG alone (protocol 3). In addition, on day 14 of the cycle, performing the OPU session in the non-stimulated control (protocol 1) and L-GF^equina^ priming before eCG injection (protocol 2) showed that L-GF^equina^ priming before eCG injection significantly (*p* < 0.05) increased the total number of follicles, the number of 5–6 mm and >7 mm diameter follicles, the number of COCs, and the recovery rate and blastocyst production when compared with non-superstimulated donors (protocol 1).

Overall, L-GF^equina^ priming before eCG injection supports the continuous development of follicles throughout the estrous cycle, increases the number and size of medium- and large-sizes follicles, the number of COCs, the recovery rate, and blastocyst production when compared with non-superstimulated (protocol 1) or eCG-superstimulated donors (protocol 3).

## 4. Discussion

In the present study, we developed a new method to optimize OPU donors for IVEP efficiency in Holstein cows. This protocol is based on priming with L-GF^equina^ before gonadotropin superstimulation followed by OPU and IVEP. To the best of our knowledge, this is the first study on intraovarian injection of L-GF^equina^ before OPU-IVEP in superstimulated Holstein cows.

In the current study, intraovarian injection of L-GF^equina^ prior to ovarian superstimulation (protocol 2) with eCG increased the number of OPU sessions per cycle, the number of COCs recovered/cow/session, the recovery rate, and the number of blastocysts/cow/session when compared to cows superstimulated with eCG alone (protocol 3) or no eCG superstimulation (protocol 1). These results may indicate that L-GF^equina^ can support continuous follicular development throughout the estrous cycle in eCG-superstimulated Holstein cows. Similarly, PRP supports follicular growth and the development of human primordial and primary follicles to the preantral stage [23]. In addition, treatment of autografts of frozen–thawed human ovarian tissue with PRP has led to live births [24]. The positive effect of PRP on follicular growth might be due to the presence of higher concentrations of growth factors that support the early stage of follicle development. In contrast, other studies reported that PRP had no effect on mouse preantral follicular growth, but it improved the follicle survival rate [25,26]. The difference between our results and those of these studies is probably due to the different supplements used (lyophilized PRP versus platelet lysate) or due to the source of PRP. In previously conducted side-by-side comparisons, there have been instances where using pFSH has produced more gratifying and predictable outcomes than using eCG. A cow from protocol 3 in our investigation was excluded because there was no follicular growth following eCG superstimulation. However, the benefits of eCG are that it may be provided in a single dose, is inexpensive, and is available in large numbers as opposed to the many injections needed when utilizing pituitary preparations. Due to its greater sialic acid concentration, eCG has a substantially longer biological half-life than pituitary FSH [27]. eCG has been found in cattle up to 10 days following injection in quantifiable concentrations [28]. Thus, in our study, the cows receiving eCG presented satisfactory follicular growth up until day 10 of the cycle (day 8 following eCG delivery).

Furthermore, in the current study, superstimulation with eCG alone or in L-GF^equina^-primed cows resulted in more follicular growth, COCs, and an increased COC recovery rate when compared with non-stimulated cows. It is well-known that eCG might act by supporting the growth of follicles in superovulation programs. The enhancing effect of eCG on follicle growth has been documented during superovulation [29] and estrous synchronization programs in cattle [30]. In contrast, ovarian follicle superstimulation with p-FSH was not observed to improve oocyte quality or IVEP compared to no hormonal treatment [30]. FSH treatment may be used to increase the numbers of follicles in the correct size categories and even the growth phase but still may not provide an ideal environment for the oocyte to acquire developmental competence [31,32,33]. This discrepancy could be due to the source, the batch of gonadotropins used, or the synchronization program. Moreover, in our study, the number of the large follicles and the recovery rate of COCs were similar in protocol 1 on days 6 and 14 compared with protocol 3 on day 10.

The results of our study are in accordance with what has been previously reported by Sills et al. [34] in humans. Furthermore, intraovarian injection of autologous PRP before superovulation resulted in higher follicular growth and blastocyst development in vivo in Holstein cows [14]. The mechanism by which L-GF^equina^ could stimulate follicular growth and blastocyst development is not entirely clear. The PRP from L-GF^equina^, through its high cytokines and growth factors, could stimulate angiogenesis and then ovarian perfusion, as well as improving the oocyte’s competence in humans [34]. In humans, it was observed that an in vitro culture of primordial follicles in the presence of PRP significantly increased follicular growth and viability when compared with a control culture [35]. Growth factor ligands interact with receptors after intraovarian injection of PRP to affect cell differentiation. Additionally, it has been proposed that intraovarian PRP therapy could improve ovarian function by boosting the number and quality of follicles by activating the ovary’s dormant follicles and encouraging the differentiation of prospective ovarian stem cells into fresh, youthful oocytes [36]. By restoring the ovarian milieu, platelet-derived growth factors may encourage follicular expansion and ovarian angiogenesis [37]. PRP is injected intraovarian to stimulate the release of a variety of growth factors, including VEGF, bFGF, PDGF, TGF, EGF, IGF, and several interleukins, as well as several other factors that may promote or trigger angiogenesis [12,38].

## 5. Conclusions

Intraovarian injection of L-GF^equina^ increased the number of OPU sessions per cycle, the development of medium- and large-sized follicles, the number of COCs/cow/session, and the blastocysts production during OPU-IVEP in Holstein cows.

## Figures and Tables

**Figure 1 animals-12-02618-f001:**
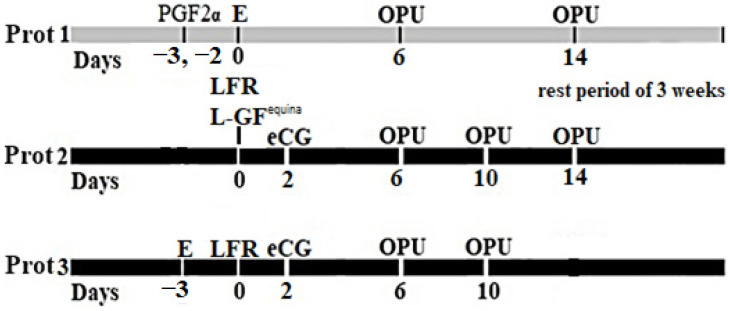
Protocol for OPU in Holstein cows. Protocol 1: estrus detection and OPU session on days 6 and 14 after its detection. Protocol 2: Large follicle (>7 mm) removal (LFR); LGF^equina^ injection on day 0; eCG injection after two days; and OPU sessions on days 6, 10, and 14 after L-GF^equina^ injection. Protocol 3: LFR on day 0; eCG injection after two days; OPU sessions on days 6 and 10 after LFR. E—estrus; PGF2α—prostaglandin; F2 alpha i.m. injections; eCG—equine chorionic gonadotrophin i.m. injections; LFR—large follicles removal; L-GF^equina^—reconstituted Lyophilized growth-promoting factor extracted from horse blood platelets.

**Figure 2 animals-12-02618-f002:**
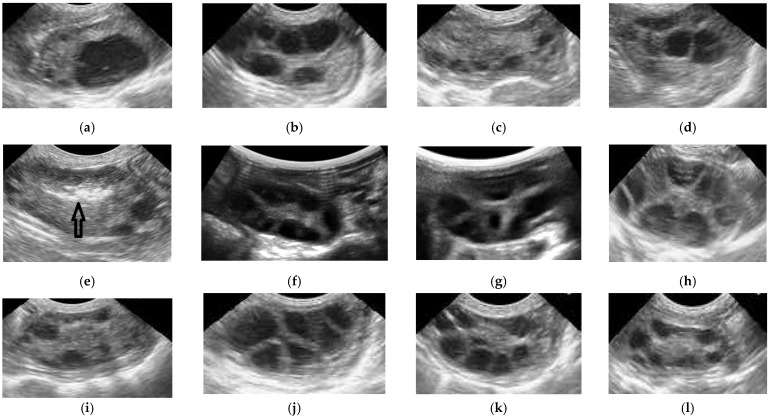
Ultrasound images representing the follicular growth during all three protocols. (**a**) Prot 1. Day 0—estrus; (**b**) Prot 1. Day 6—OPU; (**c**) Prot 1. Day 10—small follicles; (**d**) Prot 1. Day 14—OPU; (**e**) Prot 2. Day 0—LFR + L-GF^equina^ inj. (the L-GF^equina^ solution appears hyperechogenic, arrows, Day 0, in the ovary); (**f**) Prot 2. Day 6—OPU; (**g**) Prot 2. Day 10—OPU; (**h**) Prot 2. Day 14—OPU; (**i**) Prot 3. Day 0—LFR; (**j**) Prot 3. Day 6—OPU; (**k**) Prot 3. Day 10—OPU; (**l**) Prot 3. Day 14—small follicles.

**Table 1 animals-12-02618-t001:** Effect of OPU protocol on follicular development, recovery of COCs, and number of blastocysts/OPU session in Holstein cows (Mean ± SEM).

Items	Day 6			Day 10		Day 14	
	Prot. 1 (*n* = 6)	Prot. 2 (*n* = 6)	Prot. 3 (*n* = 5)	Prot. 2 (*n* = 6)	Prot. 3 (*n* = 5)	Prot. 1 (*n* = 6)	Prot. 2 (*n* = 6)
Total no. follicles	5.5 ± 0.7 ^b^	20.2 ± 0.7 ^a^	18.0 ± 1.7 ^a^	11.5 ± 1.3 ^a^	6.0 ± 0.3 ^b^	7.2 ± 0.4 ^b^	13.8 ± 1.2 ^a^
Follicles 3–4.9 mm	2.3 ± 0.2 ^c^	4.8 ± 0.4 ^b^	6.6 ± 0.9 ^a^	4.2 ± 0.3 ^a^	3.4 ± 0.2 ^a^	3.7 ± 0.2 ^a^	4.0 ± 0.4 ^a^
Follicles 5–6.9 mm	1.2 ± 0.2 ^b^	4.7 ± 0.3 ^a^	4.2 ± 0.3 ^a^	3.0 ± 0.5 ^a^	1.6 ± 0.2 ^b^	1.7 ± 0.2 ^b^	4.3 ± 0.3 ^a^
>7 mm follicles	2.0 ± 0.4 ^c^	10.7 ± 0.3 ^a^	7.2 ± 0.8 ^b^	4.3 ± 1.0 ^a^	1.0 ± 0.3 ^b^	1.8 ± 0.3 ^b^	5.5 ± 0.9 ^a^
No COCs recovered	2.0 ± 0.3 ^c^	7.7 ± 0.7 ^b^	9.4 ± 0.5 ^a^	7.7 ± 0.4 ^a^	1.6 ± 0.2 ^b^	1.8 ± 0.3 ^b^	6.8 ± 0.7 ^a^
Recovery rate	35.5 ± 4.6 ^b^	38.1 ± 3.2 ^b^	53.6 ± 8.4 ^a^	70.6 ± 8.4 ^a^	26.4 ± 3.4 ^b^	25.4 ± 3.8 ^b^	51.8 ± 7.8 ^a^
Number of blastocyst/OPU session	0.7 ± 0.2 ^b^	3.2 ± 0.3 ^a^	3.4 ± 0.4 ^a^	3.2 ± 0.4 ^a^	0.6 ± 0.2 ^b^	0.5 ± 0.2 ^b^	3.5 ± 0.2 ^a^

^a,b,c^ Superscripts within the same row indicate significant differences at *p* < 0.05 (a > b > c).

## Data Availability

The data presented in this study are available on request from the corresponding author.

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
