# Peer review of "Intraovarian Injection of Reconstituted Lyophilized Growth-Promoting Factor Extracted from Horse Blood Platelets (L-GFequina) Increases Oocytes Recovery and In Vitro Embryo Production in Holstein Cows"

_animals, 2022, doi:10.3390/ani12192618_

Round 1
Reviewer 1 Report
The article is devoted to increasing the efficiency of the in vitro system of obtaining bovine embryos using biologically active substances. The data obtained during the study are unique, since the effect of the L-GF (equina) on the development of cow follicles, cumulus-oocyte complexes (COCs) recovery rate by OPU and blastocyst development are presented.
However, there are some inaccuracies.
1. When selecting the cows used in the study, was attention paid to how long they had passed after calving?
2.It is not clear whether the researchers made the L-GF themselves or purchased it.
3. Whether one batch of L-GF was used or not?
4. Maybe you can provide more information about L-GF activity: specify concentration of platelets or growth factors activity.
5. Please provide more information about how large follicles were removed.
6. The authors indicate "large follicles (>7 mm)" (138; 144 line), while the explanations under Figure 1 state "Large follicles (> 8 mm)". Please specify.
7. How was the L-GF intraovarian procedure performed?
8. Explain why the estrus of cows was not detected in Prot2?
9. Whether the OPU procedures performed on the cows according to Prot1 could not affect the results of the experiment according to Prot2? Maybe you could provide arguments.
10. According to what criteria was the quality of oocytes evaluated?
11. How the "recovery rate" was calculated?
12. The study does not contain data on the qualitative indicators of oocytes in different research groups, their maturation, fertilization and further development. Only average data per OPU session is presented. It is difficult to assess whether the entire IVP system worked efficiently and stably throughout the study period. Maybe the authors could supplement the article with these data.
13. Maybe the authors could explain why after OPU on day6, many of the indicators analyzed in the Prot2 group do not differ or are lower (except for large follicles) than the Prot3 group.
The obtained research results are interesting and raise many questions and require more detailed and comprehensive research in the future.
Reviewer 2 Report
General comments:
The manuscript entitled "Intraovarian injection of reconstituted Lyophilized growth promoting factor extracted from horse blood platelets (L-GFequina) increase oocytes recovery and in vitro embryo production in Holstein cows” (Manuscript number; animals-1909785) was describes the effect of Lyophilized growth promoting factor extracted from horse blood platelets (L-GFequina) via intraovarian injection to cows on the reaction of OPU-IVEP results. From results, intraovarian injection of L-GFequina significantly increase OPU sessions per cycle, COC recovery, and blastocysts production. And the higher efficiency of OPU-IVEP in L-GFequina was caused by higher development of medium and large follicles. This manuscript is very interesting and provides useful information for the applied reproductive technology in dairy cows. There are some points needing corrections in the manuscript. Please consider the suggested edits listed below.
Specific Comments:
2. Materials and Methods
2.2. Experimental animals
L105-112:
Were these animals were lactating cows? Please clarified it. If they were milked, please describe the range of milk production per day during experimental period.
2.4. Assignment of Groups and Protocol of Administration
L139:
Please describe the procedure of intraovarian injection of L-GFequina in detail. Especially, the injection part of L-GFequina.
L130-148:
From your protocol, all cows which were used in Prot 1 were used in Prot 2, however, cows used in Prot 3 were not used in Prot 1. Why did you use cows repeatedly in Prot 2, but not in Prot 3? Because higher efficiency of OPU-IVEP in Prot 2 (intraovarian injection of L-GFequina) may be caused by carry over effect of previous OPU stimulation. Please answer about this idea, or if you have any background that previous OPU did not effect the efficiency of OPU-IVEP.
In Prot 2, please describe the start timing of LFR. Did you start LFR after 2 or 3 days after estrus as same as Prot 3? If so, did you induce estrus by hormonal treatment? If you did not concern about the estrus timing in Prot 2, the endocrine status may be different from Prot 3; for example, if you did not confirmed estrus, some cows used in Prot 2 might have corpus luteum, so plasma progesterone concentration of these cows was higher in cows used in Prot 2 than these in Prot 3. Or did you confermed corpus luteum in Prot 3? Please clarified it. In addition, considering of the protocol of Prot 1, all cows in Prot 1 have corpus luteum through the experiment, so the plasma progesterone in Prot 1 were higher than that in Prot 2 and Prot 3. I wonder this endocrine status may affect the follicle development among treatment groups and may affect the efficiency of OPU-IVEP results. Because the level of plasma progesterone concentration affect the follicle growth rate and pattern of follicle development. What do you think? Please answer these questions.
In Prot 3, please clarified whether estrus was spontaneous or induced by hormonal treatment. If estrus induction was different from Prot 1, please describe the reason of it.
Please describe the criteria of implement of OPU-IVEP. You did OPU-IVEP on days 6 and 14 in Prot 1, however you did OPU-IVEP on days 6, 10, and 14 in Prot 2 and on days 6 and 10 in Prot 3. Why you did OPU-IVEP in different protocol in each protocol. Please clarify these differences.
L154:
“Large follicles (> 8 mm) removal” to “Large follicles (> 7 mm) removal”?
L283
“ECG” to “eCG”
Reviewer 3 Report
This is a mostly technical but very useful article showing a novel protocol to induce follicle development in cattle.
I hace only two question/concern Table 1 shows that COC recovered with protocol, 1 (after 6 day) are 2.0+/- 0.3 but at 10 day with protocol 3 are 1.6 +/- 0.2 and 1.8 +/- 0.3 with protocol 1. I understood since they have different supperscripts (from protocol 1 at day 6) they are different. But from a biological point of view those number means that in all three grupos you get 1-3 COC and that means no differences. The I think the author could mention that.
The same with the rwo of 7mm oocytes
The second question is you evsluate the time to reach the blastocyst stage in the different goups Where they similar?
Round 2
Reviewer 1 Report
The article has been partially corrected according to comments.
Author Response
Thank you for the constructive comments on our manuscript entitled “Intraovarian injection of reconstituted Lyophilized growth-promoting factor extracted from horse blood platelets (L-GFequina) increase oocytes recovery and in vitro embryo production in Holstein cows”
Reviewer 2 Report
Thank you for responding for my comments and revised your manuscript.
I have some another comments for your manuscript
1. Please more describe precisely about the injection point of L-GFequina. Middle of the ovary? Or surface of ovary?
2. You mentioned that;
"Yes, in Prot 1 all the cows presented corpus luteum after ovulation but we expected at this and we wanted to show how much COCs can be recovered during an estrus cycle. We didn't expect to have more recovery COCs in Prot 1 than Prot 2, because the plasma progesterone concetration affects the growth of the follicles. "
If so, I think you should not compare Prot1 and Prot2 results. Because, you could not excluded the possibilities of lower COCs recovery rate in Prot1 than Prot2 caused by higher plasma progesterone concentration. Therefore, if you want to state the advantageous of the effect of L-GFequina, you should start OPU 6 days after estrus as same as Prot1 which bearing corpus luteum. Or you should just show the results of Prot2 and Prot 3.
3. You mentioned that;
"In this study, the OPU sessions were carried out on days 6 and 14 for Protocol 1, days 6, 10, and 14 for Protocol 2, and days 6 and 10 for Protocol 3 by the presence of numerous follicles that can be punctured (Figure 2). In other cows, we used different times interval and the results were not too good. For example when we proceed OPU two days after eCG the number of recovered COCs was 2 or 3. When we expected two days more we recovered more than 7 COCs. From another point of view when we proceed with OPU 7 days or more after eCG, even if the ovary shows a lot of medium and large follicles we didn't recover any COCs or one, two with obvious signs of atresia and a black-punctate cytoplasm. "
From this results, Did L-GFequina increased the OPU sessions caused by higher follicles development? In other words, This treatment increased the repeatability of OPU sessions? If so, how about add this fact in the Discussion?
Round 3
Reviewer 2 Report
Thank you for responding for my comments and revised your manuscript. I understand your opinions.